# Preliminary Sizing of Electric-Propulsion Powertrains for Concept Aircraft Designs

**Josin Hu** and **Julian Booker** *

Electrical Energy Management Research Group, Faculty of Engineering, University of Bristol, Bristol BS8 1TR, UK
* Correspondence: j.d.booker@bristol.ac.uk

**Abstract:** The drive towards a greener and more sustainable future is encouraging the aviation industry to move towards increasing electrification of its fleet. The development of electric propulsion technologies also requires new approaches to assess their viability in novel configurations. A methodology is proposed which consists of four sub-procedures; powertrain modelling, performance analysis, aerodynamic modelling, and sizing. This approach initially considers powertrain modelling using AIAA symbol representations, and a review of the available literature establishes state-of-the-art component values of efficiency, specific power, specific energy, and specific fuel consumption. The sizing procedure includes a mission and aerodynamic analysis to determine the energy and power requirements, and it relies on a mass regression model based on full-electric, hybrid, VTOL and fixed-wing aircraft found in the literature. The methodology has been applied to five case studies which are representative of a wide range of missions and configurations. Their predicted masses from the sizing procedure have been validated against their actual masses. The predicted total mass shows generally good agreement with the actual values, and in addition, accurate values for active mass have been predicted. A sensitivity analysis of the sizing procedure suggests that future work may include a more accurate analysis of aerodynamics and mission if the methodology were to be applied for selecting aircraft concepts.

**Keywords:** aircraft; electric-propulsion; powertrain; design; modelling





## 1. Introduction

Sustained growth of air passenger traffic is expected to increase by 4.5% annually [1]. However, regulation related to protecting the environment—reducing emissions, noise, and pollution—will force the industry to develop quieter and greener alternatives [1–3]. A major technological development concerns the electrification of aircraft, which is characterised by two trends: abandoning pneumatic or hydraulic systems in favour of electrical systems, and the development of novel hybrid and all-electric propulsion [1,2]. Electrification has the potential of enabling aircraft with higher fuel efficiency, lower noise, and lower environmental impact air travel [1], and it is a threat as well as an opportunity for the UK supply chain, which may benefit from entering these new market segments [1].

Nevertheless, novel propulsion technologies are in their infancy, and designers do not possess mature methods yet to consider the impact of electrification on their designs. Therefore, with a focus on propulsion, this paper aims to establish methods for exploring the newly opened-up design space, enabling the designer to compare different aircraft configurations. Not only will this progress the necessary shift to greener alternatives, but increased electrification will also open up new market segments such as the development of Vertical Take-off and Landing (VTOL) platforms for urban taxi or air transport applications [1,2].

### 1.1. Background

Part of the drive for electrification is the gradual development of More Electric Aircraft (MEA). MEA still use conventional fuels for propulsion but have increasingly electrified

onboard systems [2]. Examples include actuation, de-icing, and air-conditioning, which were formerly pneumatic, mechanical, or hydraulic [2]. MEAs benefit from the improved reliability, efficiency, and maintainability related to electrical systems [1,2]. Nevertheless, advances in power electronics are required to sustain this development, and it is expected to unfold gradually [1,2].

Electrified propulsion on the other hand is a disruptive development and allows for radical increases in efficiency and emissions reduction, which the gradual evolutionary trend of MEA cannot deliver [2,4]. Almost 100 electric-propulsion concepts are already in development [3] such as the EHang AAV [5], Kitty Hawk [6], Bartini [7], and Airbus Vahana [8]. There exists a variety of electrical-propulsion configurations—the same configurations as used by the automotive industry [2]—characterised by the amount of electricity used as well as powertrain arrangement [4].

The first is an all-electric or full-electric powertrain, where all propulsive energy originates from batteries, fuel cells or capacitors [1,4]. Although fully benefiting from the advantages of electrification, these configurations are limited by the low specific energy of batteries compared to conventional fuels [3], meaning that they are restricted to the <20 seat sub-regional and other market segments [1].

Turbo-electric (or serial [9]) powertrains, on the other hand, generate energy by using conventional fuel but convert this chemical energy to electrical energy for propulsion [2,4]. Although less efficient than all-electric configurations, their missions are not limited, and this still enables novel concepts such as distributed propulsion to be implemented [4].

Finally, hybrid-electric powertrains are characterised by the usage of more than one type of energy source: batteries and fuel [2,4]. Through the means of an engine, the fuel is converted into mechanical energy. A further distinction may be made between series or parallel type configurations [1]. For a series-hybrid configuration, the engine drives a generator which connects with the battery at an electrical node, whereas, for a parallel-hybrid configuration, the battery drives an electrical motor which connects with the engine at a mechanical node [1,4] or not at all [10].

The main benefit of a series type configuration is the decoupling of the energy sources, meaning that the engine is run independently and can be operated at its maximum efficiency [4]. The parallel type of configuration on the other hand benefits from weight savings since fewer components are required, but also means that the operation of the electrical and fuel energy sources are coupled mechanically [4]. Furthermore, the mechanical connection itself introduces additional complexity [4], and novel propulsive technologies face additional challenges in the form of electrical machine performance and battery safety [2], as well as requiring improvements in the reliability of power electronics [1].

As the most important consideration in aircraft design, the sizing process is responsible for determining the required weight of the aircraft given a payload [11]. Traditionally, determining the total, empty, and fuel mass of the aircraft has been achieved by utilising an iterative method as used by Roskam [12] or using fuel fractions and the Breguet equation as used by Raymer [11] and is informed by the mission requirements of the given aircraft [11,12].

Other major considerations in aircraft design are the wing loading and power loading in the case of propeller aircraft or thrust loading in the case of jet aircraft, which are closely related to aircraft performance [11,12]. Conventionally, these have been chosen by a performance analysis of the aircraft (known under various names such as wing-loading to thrust-loading diagrams [12], carpet plots [11], constraints analysis [13], sizing matrix plots [14], etc.), in which a design region is constrained by performance requirements. These include consideration of stall, take-off, landing, climb, manoeuvring, etc. [12].

Traditionally, performance analysis and sizing were independent procedures [11,12]. However, due to the coupling of battery weight to performance, there is a need to integrate these two analyses [14]. Design methods have already been developed, for example, the integrated methodology presented by Riboldi and Gualdoni can be used for the design of small all-electric platforms at current technology levels [14], or small hybrid-electric air-

craft [15]. De Vries et al. [16] have developed a preliminary sizing method for hybrid-electric propulsion platforms including aero-propulsive interaction. Pornet [13] has developed conceptual design methods for sizing and performance of hybrid-electric transport aircraft, whereas Finger et al. [17] have discussed hybrid-electric general aviation aircraft. Raymer [11] too suggested using Battery Mass Fractions (BMF) as an electrical equivalent to the fuel fractions used in conventional methodologies. Geiss and Voit-Nitschmann and Jansen et al. have similarly adapted the Breguet equation for electric aircraft [18] and for turbo-electric aircraft [19] for use in conventional design processes.

*1.2. Methodology*

Unlike existing approaches proposed by researchers, this paper proposes a methodology focused on comparing different aircraft configurations. The methodology is shown in Figure 1 and its importance lies in being able to narrow down a heterogeneous design space through a unified modelling approach. It is capable of comparing fixed-wing to VTOL and rotorcraft configurations as well as considering the differences between all-electric, series-hybrid or parallel-hybrid configurations.

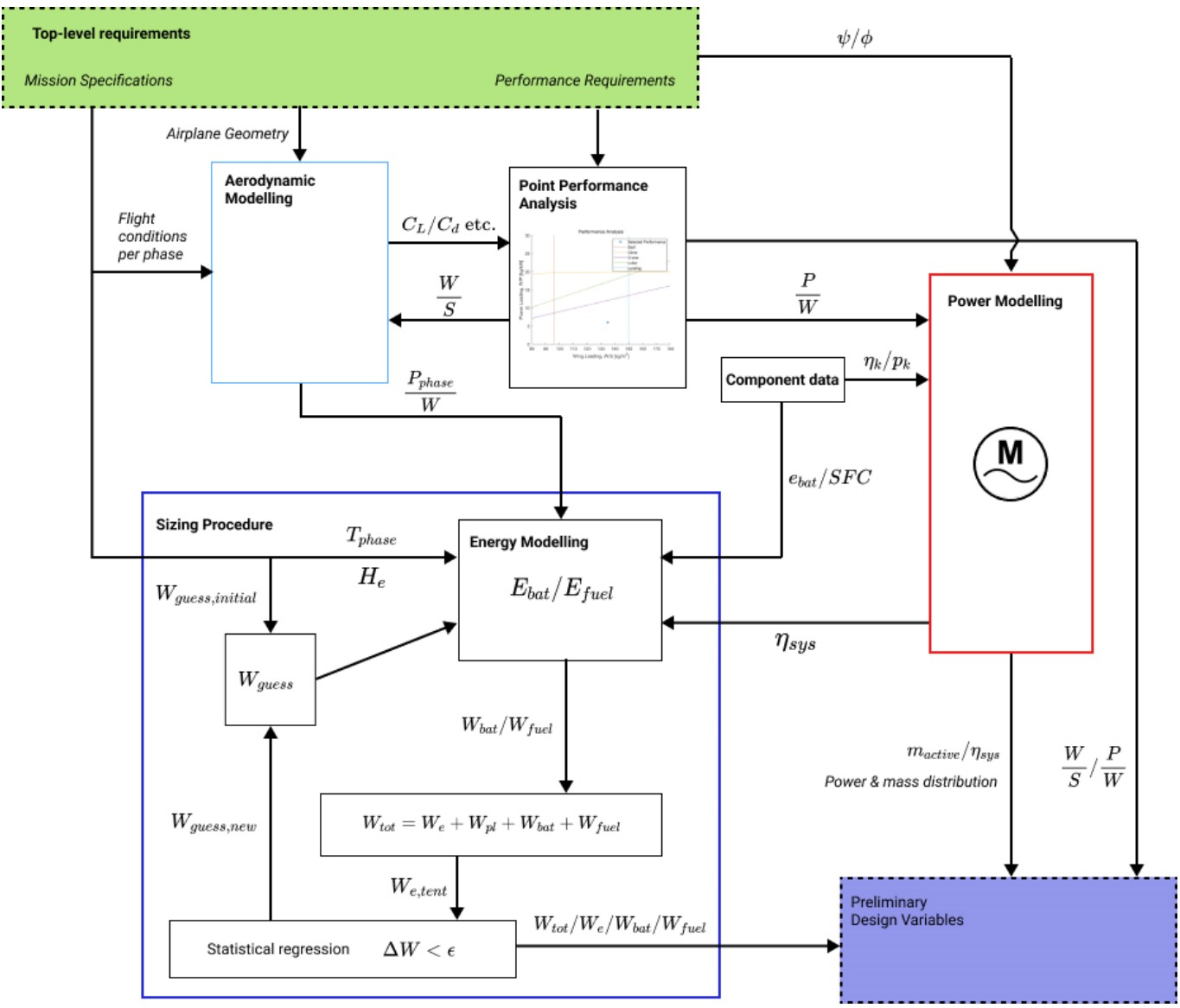

**Figure 1.** Methodology for calculating design variables.

The methodology consists of four sub-procedures: aerodynamic modelling, point performance analysis, powertrain modelling, and the sizing procedure. Comparatively, more attention will be paid to powertrain modelling through a block diagram approach using AIAA symbols, which was adapted from work done by Booker et al. [20] for considering the reliability and efficiency of a powertrain. The designer is thereby provided with a way to distinguish between and compare propulsive platforms through the use of system specific power and system efficiency, which were found to be key performance parameters by Jansen et al. [19] for turbo-electric drives.

Aerodynamic modelling and the sizing procedure were strongly inspired by the work done by Riboldi and Gualdoni [14] for small-electric aircraft. They proposed calculating the required power for each phase of flight by considering the aircraft in static-flight equilibrium, Bacchini and Cestino [6] provided an expression for the hover phase of VTOL configurations. The aerodynamic model informs the sizing process. The regression is used for finding the total and empty weight of the aircraft is based on a relationship identified for conventional aircraft by Roskam [12].

The validity of the methodology will be tested against five case studies and different aircraft types: fixed-wing general-aviation; fixed-wing full-electric glider; VTOL logistics; VTOL urban mobility (5 passengers); and VTOL urban mobility (10 passengers). The selected wing loading and power loading of these case studies is known, and therefore point performance analysis shall be entirely neglected.

## 2. Powertrain Modelling

The power train model is inspired by the block diagram method used for reliability and efficiency modelling set forth by Booker et al. [20]. Rather than model reliability and efficiency, the method has been adjusted to model power flow and efficiency. A selected number of symbols developed by the American Institute of Aeronautics and Astronautics (AIAA) [20,21] have been adapted for representing the system (Figure 2). An addition to this set of symbols is a Power Control Unit (PCU) symbol which intends to simplify the modelling of converters, rectifiers, inverters, etc. [20]. The PCU block must be connected to each instance of an electrical motor [20]. Figure 3 shows a selection of powertrains modelled using the AIAA symbols.

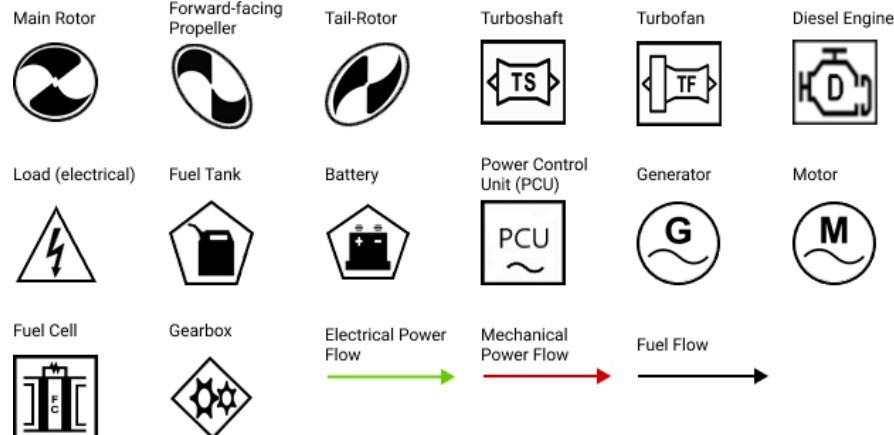

**Figure 2.** Adapted AIAA symbols. (From [21] and reprinted by permission of the American Institute of Aeronautics and Astronautics, Inc.).

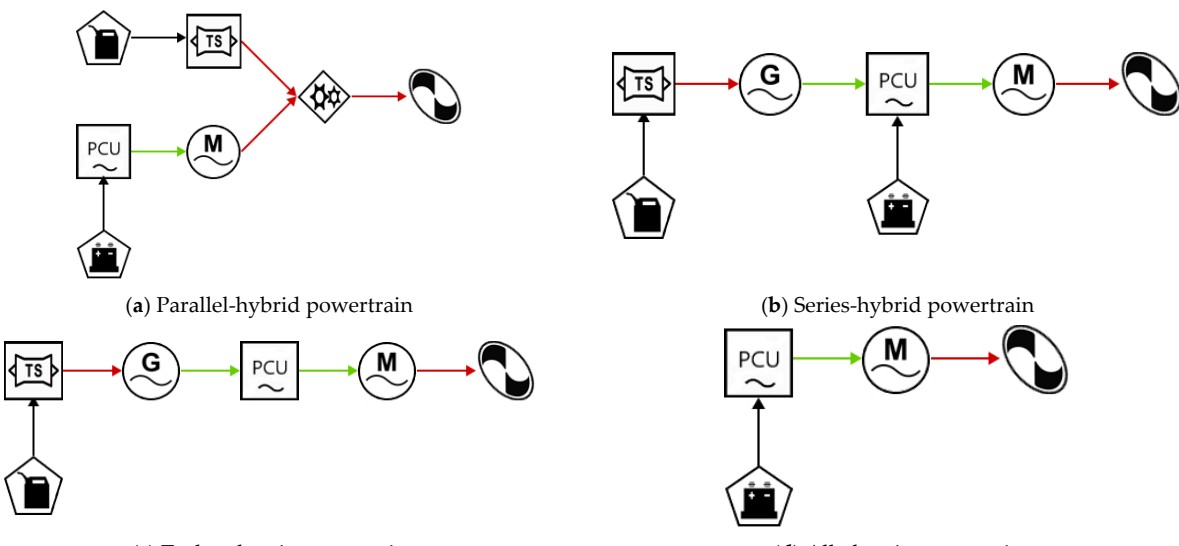

(**a**) Parallel-hybrid powertrain

(**b**) Series-hybrid powertrain

(**c**) Turbo-electric powertrain

(**d**) All-electric powertrain

**Figure 3.** Selection of powertrain architectures modelled using AIAA symbols.

### 2.1. Series and Parallel Connections

Any given powertrain is modelled as a combination of series (Figure 4) and parallel (Figure 5) connections. By considering the rules for combining or decomposing such connections, it is possible to determine the power distribution, system mass, equivalent system efficiency, and equivalent system specific power of any given powertrain.

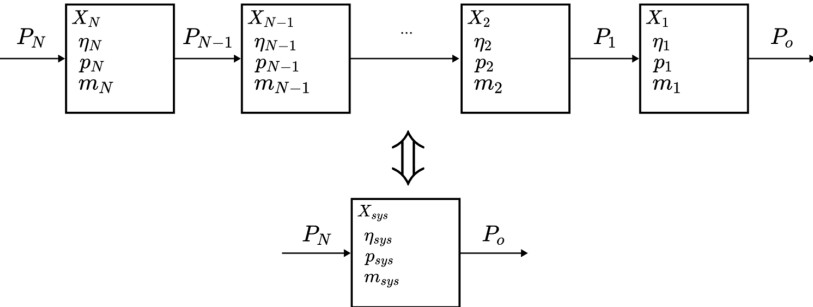

**Figure 4.** Series connection.

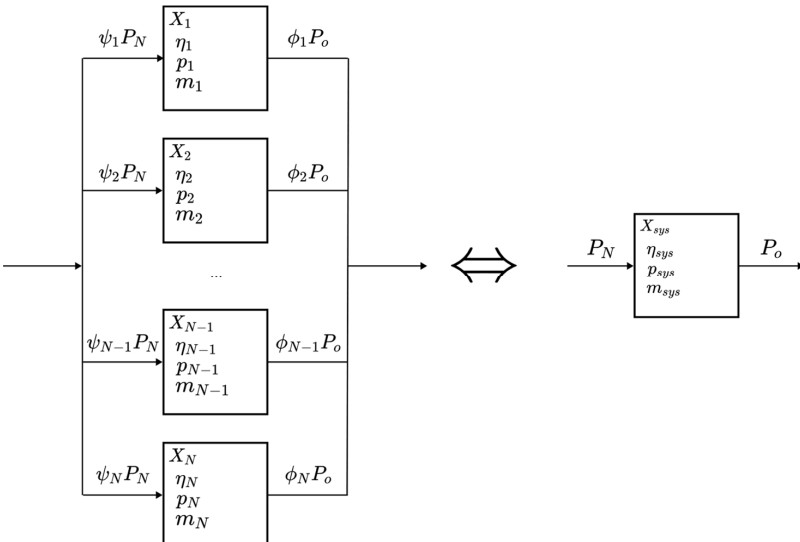

**Figure 5.** Parallel connection.

The combinatory rules for series and parallel connections follow from the definitions of efficiency, $\eta_k$, and specific power, $p_k$, of a single component $X_k$ (Figure 6):

$$\eta_k = \frac{P_o}{P_k} \tag{1}$$

$$p_k = \frac{P_k}{m_k} \tag{2}$$

where $P_o$ is the output power of the component, $P_k$ is the input power of the component, and $m_k$ is the mass of the component. The specific power is defined in terms of input power since $P_k \geq P_o$, therefore the specific power will be a more conservative approximation.

**Figure 6.** Definition of single component, $X_k$.

An exception to this convention is internal combustion engines, e.g., diesel or turboshaft types. These engines generate power and are conventionally characterised by their output power. Hence, the specific power of turboshaft and diesel engines is defined to be:

$$p_k = \frac{P_o}{m_k} \tag{3}$$

For a series connection (Figure 4), an expression for the system efficiency of a series connection may thus be derived by considering the input power, $P_N$, and output power $P_o$ of the entire connection:

$$P_o = \eta_1 \eta_2 \cdots \eta_{N-1} \eta_N P_N \Rightarrow \eta_{sys} = \eta_1 \eta_2 \cdots \eta_{N-1} \eta_N = \prod_{k=1}^{N} \eta_k \tag{4}$$

The mass of a single element within a series connection may similarly be expressed in terms of the output power, $P_o$, or alternatively input power, $P_N$, of the connection and its specific power, $p_k$:

$$m_k = \frac{P_o}{p_k \eta_1 \eta_2 \cdots \eta_k} = \frac{P_o}{p_k \prod_{i=1}^{k} \eta_i} = \frac{\eta_{sys} P_N}{p_k \prod_{i=1}^{k} \eta_i} \tag{5}$$

The series system mass, $m_{sys}$, and equivalent system specific power, $p_{sys}$, may be derived to be:

$$m_{sys} = \frac{P_N}{p_N} + \frac{\eta_N P_N}{p_{N-1}} + \cdots + \frac{\eta_N \cdots \eta_2 P_N}{p_1} = \sum_{k=1}^{N} \frac{\eta_{sys} P_N}{p_k \prod_{i=1}^{k} \eta_i} \Rightarrow \frac{1}{p_{sys}} = \sum_{k=1}^{N} \frac{\eta_{sys}}{p_k \prod_{i=1}^{k} \eta_i} \tag{6}$$

When considering a parallel connection (Figure 5), two additional parameters must be defined: an input parameter $\psi_k$, which is defined as the ratio of component input power to the total system input power, and an output parameter $\phi_k$, which similarly is defined as the ratio of component output power to the total system output power. These parameters must satisfy the following relationships:

$$\phi_1 + \phi_2 + \cdots + \phi_N = \sum_{k=1}^{N} \phi_k = 1 \tag{7}$$

$$\psi_1 + \psi_2 + \cdots + \psi_N = \sum_{k=1}^{N} \psi_k = 1 \tag{8}$$

Conceptually, these two parameters represent the power distribution of the system and describe the amount of power flowing through each branch of a parallel connection and they are closely related to the mission which determines how much power comes from conventional fuel sources and how much power comes from electrical fuel sources.

The system efficiency can be found by again considering the relationship between input and output power:

$$P_o = \eta_1 \psi_1 P_N + \eta_2 \psi_2 P_N + \cdots + \eta_N \psi_N P_N = P_N \sum_{k=1}^{N} \psi_k \eta_k \Rightarrow \eta_{sys} = \sum_{k=1}^{N} \psi_k \eta_k \tag{9}$$

$$P_N = \phi_1 P_o \eta_1 + \phi_2 P_o \eta_2 + \cdots + \phi_N P_o \eta_N = P_o \sum_{k=1}^{N} \frac{\phi_k}{\eta k} \Rightarrow \frac{1}{\eta_{sys}} = \sum_{k=1}^{N} \frac{\phi_k}{\eta k} \tag{10}$$

There are two ways of calculating system efficiency, either in terms of the input parameter (9) or output parameter (10). It is interesting to note how the system efficiency depends on the power distribution; maximising system efficiency involves selecting the most efficient components but also routing power through the more efficient components.

The component mass, system mass, and system equivalent specific power can also be expressed in terms of the input parameter or output parameter:

$$m_k = \frac{\phi_k P_o}{p_k \eta_k} = \frac{\psi_k P_N}{p_k} \Rightarrow m_{sys} = \sum_{k=1}^{N} \frac{\phi_k P_o}{p_k \eta_k} = \sum_{k=1}^{N} \frac{\psi_k P_N}{p_k} \Rightarrow \frac{1}{p_{sys}} = \sum_{k=1}^{N} \frac{\phi_k \eta_{sys}}{p_k \eta_k} = \sum_{k=1}^{N} \frac{\psi_k}{p_k} \tag{11}$$

### 2.2. State-of-the-Art Component Values

From a review of the literature, state-of-the-art values of efficiency and specific power were collated for each of the components (Table 1). Future values, where found, have been included as well for three timeframes: near-term (2025), mid-term (2030), and long-term (2030+). It is expected that radical advances in superconducting technology play a key role in enabling higher power densities [13,22]. From this, maximum and minimum values found in the literature have been collated, and sample mean, sample median, and sample variance derived from these values have been calculated and included as well (see Tables 2 and 3).

It was assumed that all gearboxes, shafting, cabling, and propeller have negligible mass and were not listed in Table 3. Batteries and fuel too are considered to be of negligible mass during power modelling since their mass is determined during the sizing process and depends on the mission. Battery mass and fuel mass are not included as part of the powertrain mass. Negligible mass may be modelled as infinite specific power. Similarly, fuel is considered to be perfect power transfer and may be modelled as efficiency at 100%. Note that fuel is considered to be a separate component from combustion engines such as the turboshaft or diesel engine.

**Table 1.** Efficiency and specific power values found from literature.

| Component | Efficiency Values | | | | Specific Power Values | | | |
|---|---|---|---|---|---|---|---|---|
| Main Rotor Gearbox | 0.96 [23] | 0.94 [20] | | | | | | |
| Bevel Gearbox | 0.96 [23] | 0.93 [24] | 0.99 [24] | 0.97 [20] | | | | |
| Multiplier/Reducer Gearbox | 0.96 [23] | 0.94 [24] | 0.98 [24] | 0.94 [20] | | | | |
| Shafting | 0.99 [20] | | | | | | | |
| Power Cables | 0.99 [25] | 0.99 [20] | | | | | | |
| Mid-term | 0.99 [13] | 1.00 [26] | 1.00 [25] | 1.00 [27] | | | | |
| Power Control Unit (PCU) | 0.95 [28] | 0.97 [13] | 0.95 [25] | 0.95 [1] | 8.2 [28] | 16.4 [28] | 11.0 [13] | 11.0 [22] |
| | 0.97 [20] | | | | 2.0 [1] | 4.0 [1] | | |
| Near-term | 0.97 [16] | 0.97 [1] | 0.98 [20] | | 13.0 [16] | 7.5 [25] | 10.0 [1] | |
| Mid-term | 0.99 [16] | 0.99 [29] | 0.993 [29] | 1.00 [13] | 19.0 [16] | 19.0 [29] | 26.0 [29] | 24.7 [28] |
| | 1.00 [26] | 0.98 [1] | 0.99 [27] | 1.00 [27] | 18.0 [13] | 25.0 [13] | 20.0 [22] | 20.0 [26] |
| | 0.99 [23] | | | | 17.0 [1] | 31.0 [27] | 49.0 [27] | |
| Long-term | 1.00 [16] | 0.99 [28] | 1.00 [28] | 0.98 [25] | 32.0 [16] | 16.4 [28] | 32.8 [28] | 15.0 [25] |
| | 0.98 [1] | 0.99 [27] | | | 25.0 [1] | 25.0 [27] | | |
| Turboshaft (200 kW) | 0.30 [23] | 0.20 [20] | 0.30 [20] | | 3.1 [30] | 1.2 [10] | | |
| Fuel Cell | 0.65 [20] | | | | 0.7 [13] | | | |
| Mid-term | 0.55 [13] | 0.60 [27] | 0.83 [20] | | | | | |
| Long-term | | | | | 1.0 [13] | 5.0 [27] | | |
| Motor/Generator | 0.95 [31] | 0.95 [13] | 0.95 [25] | 0.90 [1] | 5.0 [13] | 3.0 [1] | 5.0 [10] | |
| | 0.92 [20] | | | | | | | |
| Near-term | 0.92 [16] | 0.93 [1] | | | 9.0 [16] | 7.5 [25] | 7.5 [1] | |
| Mid-term | 0.96 [16] | 0.96 [29] | 0.98 [29] | 0.96 [1] | 13.0 [16] | 13.0 [29] | 16.0 [29] | 15.0 [13] |
| | 0.96 [27] | 0.96 [23] | 0.99 [20] | | 20.0 [13] | 8.0 [22] | 25.0 [22] | 20.0 [26] |
| | | | | | 12.0 [1] | 21.0 [27] | 7.7 [23] | |
| Long-term | 0.99 [16] | 0.99 [28] | 1.00 [28] | 0.99 [13] | 22.0 [16] | 15.0 [25] | 20.0 [1] | 19.0 [27] |
| | 1.00 [26] | 0.98 [25] | 0.96 [1] | 0.98 [27] | 25.0 [27] | | | |
| | 0.99 [27] | | | | | | | |
| Diesel Engines | 0.40 [30] | 0.40 [20] | | | 0.8 [30] | 4.2 [30] | | |
| Battery | 0.93 [25] | 0.90 [20] | 0.70 [32] | 1.00 [32] | 0.7 [6] | 1.3 [6] | 2.0 [13] | 2.0 [33] |
| | | | | | 3.0 [1] | 0.01 [32] | 2.0 [32] | |
| Near-term | | | | | 3.0 [25] | 5.0 [1] | 7.5 [1] | |
| Mid-term | 0.99 [34] | 0.90 [26] | 0.95 [25] | 0.60 [27] | | | | |
| | 0.99 [20] | | | | | | | |
| Long-term | | | | | 0.3 [35] | 0.4 [22] | 0.6 [22] | 6.0 [25] |
| | | | | | 7.5 [1] | 10.0 [1] | 1.0 [23] | |

**Table 2.** Compiled values of state-of-the-art component efficiency, $\eta_X$ [-].

| Component | Minimum | Maximum | Sample Mean | Sample Median | Sample Variance ($\times 10^{-3}$) |
|---|---|---|---|---|---|
| Main Rotor Gearbox | 0.940 | 0.960 | 0.950 | 0.950 | 0.200 |
| Bevel Gearbox | 0.930 | 0.990 | 0.963 | 0.965 | 0.625 |
| Multiplier/Reducer Gearbox | 0.940 | 0.980 | 0.955 | 0.950 | 0.367 |
| Shafting | 0.990 | 0.990 | 0.990 | 0.990 | - |
| Power Cables | 0.990 | 0.990 | 0.990 | 0.990 | 0.000 |
| Mid-term | 0.990 | 1.000 | 0.996 | 0.996 | 0.170 |
| Propeller | 0.870 | 0.870 | 0.870 | 0.870 | - |
| Power Control Unit (PCU) | 0.950 | 0.970 | 0.958 | 0.950 | 0.120 |
| Near-term | 0.970 | 0.980 | 0.973 | 0.970 | 0.033 |
| Mid-term | 0.980 | 0.995 | 0.991 | 0.990 | 0.022 |
| Long-term | 0.980 | 1.000 | 0.989 | 0.989 | 0.064 |
| Turboshaft (200 kW) | 0.195 | 0.300 | 0.265 | 0.300 | 3.675 |
| Fuel Cell | 0.650 | 0.650 | 0.650 | 0.650 | |
| Mid-term | 0.550 | 0.830 | 0.660 | 0.600 | 22.30 |
| Motor/Generator | 0.900 | 0.950 | 0.934 | 0.950 | 0.530 |
| Near-term | 0.920 | 0.930 | 0.925 | 0.925 | 0.050 |
| Mid-term | 0.960 | 0.990 | 0.967 | 0.960 | 0.157 |
| Long-term | 0.960 | 0.997 | 0.986 | 0.990 | 0.129 |
| Diesel Engines | 0.395 | 0.400 | 0.398 | 0.398 | 0.013 |
| Battery | 0.700 | 1.000 | 0.880 | 0.910 | 16.41 |
| Mid-term | 0.600 | 0.990 | 0.890 | 0.950 | 26.93 |

**Table 3.** Compiled values of state-of-the-art component specific power, $p_X$ [kW/kg].

| Component | Minimum | Maximum | Sample Mean | Sample Median | Sample Variance |
|---|---|---|---|---|---|
| Power Control Unit (PCU) | 2.00 | 16.40 | 8.77 | 9.60 | 27.41 |
| Near-term | 7.50 | 13.00 | 10.17 | 10.00 | 7.58 |
| Mid-term | 17.00 | 49.00 | 24.43 | 20.00 | 84.45 |
| Long-term | 15.00 | 32.80 | 24.37 | 25.00 | 56.28 |
| Turboshaft | 1.18 | 3.12 | 2.15 | 2.15 | 1.88 |
| Fuel Cell | 0.71 | 0.71 | 0.71 | 0.71 | 0.00 |
| Long-term | 1.00 | 5.00 | 3.00 | 3.00 | 8.00 |
| Motor/Generator | 3.00 | 5.00 | 4.33 | 5.00 | 1.33 |
| Near-term | 7.50 | 9.00 | 8.00 | 7.50 | 0.75 |
| Mid-term | 7.70 | 25.00 | 15.52 | 15.00 | 30.33 |
| Long-term | 15.00 | 25.00 | 20.20 | 20.00 | 13.70 |
| Diesel Engines | 0.8. | 4.15 | 2.49 | 2.49 | 5.51 |
| Battery | 0.01 | 3.00 | 1.57 | 2.00 | 0.98 |
| Near-term | 3.00 | 7.50 | 5.17 | 5.00 | 5.08 |
| Long-term | 0.30 | 10.00 | 3.69 | 1.00 | 16.42 |

*2.3. Assumption of Constant Efficiency and Specific Power*

For power modelling, it was assumed that efficiency and specific power are the same for all component sizes across all operating ranges. One consequence of this assumption is that for a parallel system, any branch containing the same elements may be equivalently represented as a single branch. Consider for example a system with two branches (Figure 7). Then, if the efficiency of both branches is equal:

$$\eta = \eta_A = \eta_B \Rightarrow \eta_{sys} = \phi_1\eta + \phi_2\eta = \eta \tag{12}$$

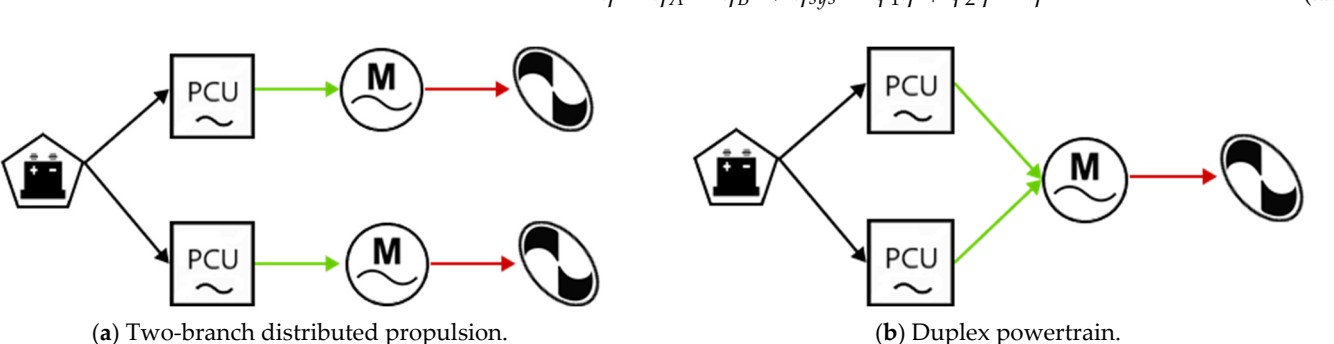

(**a**) Two-branch distributed propulsion.    (**b**) Duplex powertrain.

**Figure 7.** Examples of a parallel connection with equivalent branches.

Similarly, if the specific power of both branches is equal:

$$p = p_1 = p_2 \Rightarrow \frac{1}{p_{sys}} = \sum_{k=1}^{N} \frac{\phi_k \eta_{sys}}{p_k \eta_k} = \frac{1}{p} \tag{13}$$

Thus, the distributed propulsion system in Figure 7a, and the duplex propulsion system in Figure 7b, are equivalent to the all-electric powertrain described in Figure 3d. Concerning multiplex engines in particular (Figure 7b), if the specific power and efficiency of the components are equal, then any multiplex configuration will be equivalently modelled to that of a simplex configuration using this method.

Although for the remainder of the paper constant efficiency and specific power are assumed, it may be beneficial to consider a case where the efficiency and specific power of a turboshaft engine are related to the power output by the following relationships: $\eta_k = \alpha P_o^\beta$ and $p_k = \gamma P_o^\delta$, where $\alpha = 0.064$, $\beta = 0.21$, $\gamma = 0.220$, and $\delta = 0.50$ (Figure 8) found by Stagliano and Lobentanzer [30]. Riboldi and Gualdoni [14] have similarly found

a non-linear model for electrical machines, and Riboldi [15] has found one for internal combustion engines.

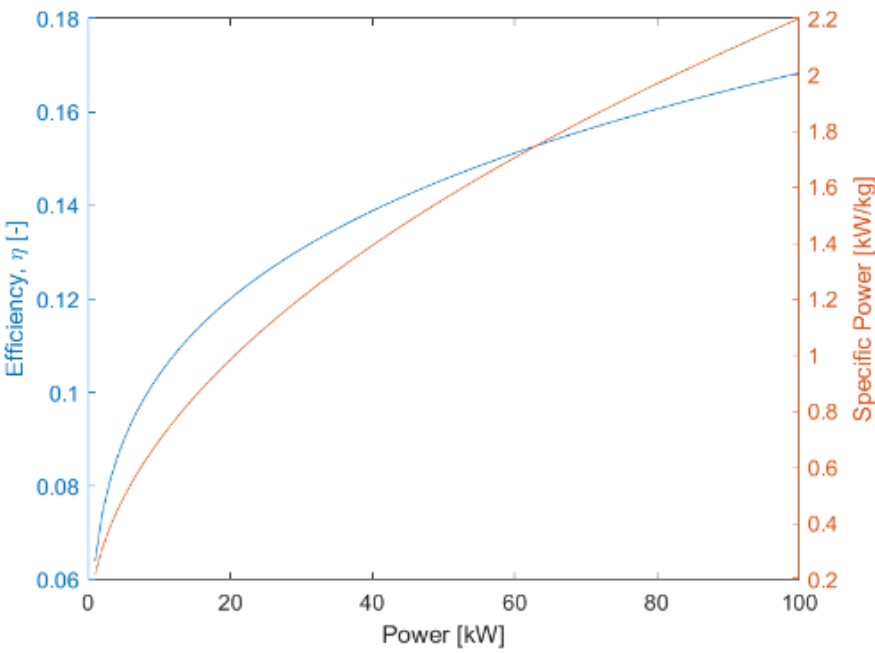

**Figure 8.** Non-linear turbo-engine surrogate model.

Hence, considering a system of just two branches: A and B (Figure 9). The power distribution is given by $\phi_A$ and $\phi_B = 1 - \phi_A$; the total system output power is given by $P_o$. Therefore, the efficiency and specific power are given by:

$$\eta_{comb} = \alpha \left( \frac{P_o}{\eta_1} \right)^\beta \left[ \frac{\eta_1}{\phi_A^{1-\beta} + \phi_B^{1-\beta}} \right] \tag{14}$$

$$m_{comb} = \frac{P_o}{\eta_1} \left[ \frac{1}{p_1} + \frac{1}{\gamma \left( \frac{P_o}{\eta_1} \right)^\delta} \left( \phi_A^{1-\delta} + \phi_B^{1-\delta} \right) \right] \tag{15}$$

where $\eta_1$ and $p_1$ are the efficiency and specific power of the generator. Figure 10 shows how the combined mass and efficiency of the system changes based on the power distribution between the two components.

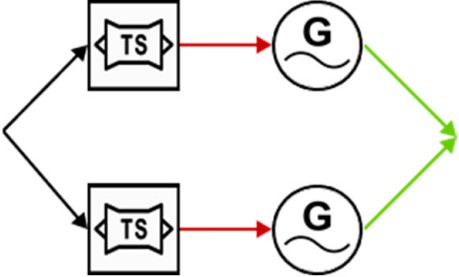

**Figure 9.** Combined system with non-constant efficiency and power.

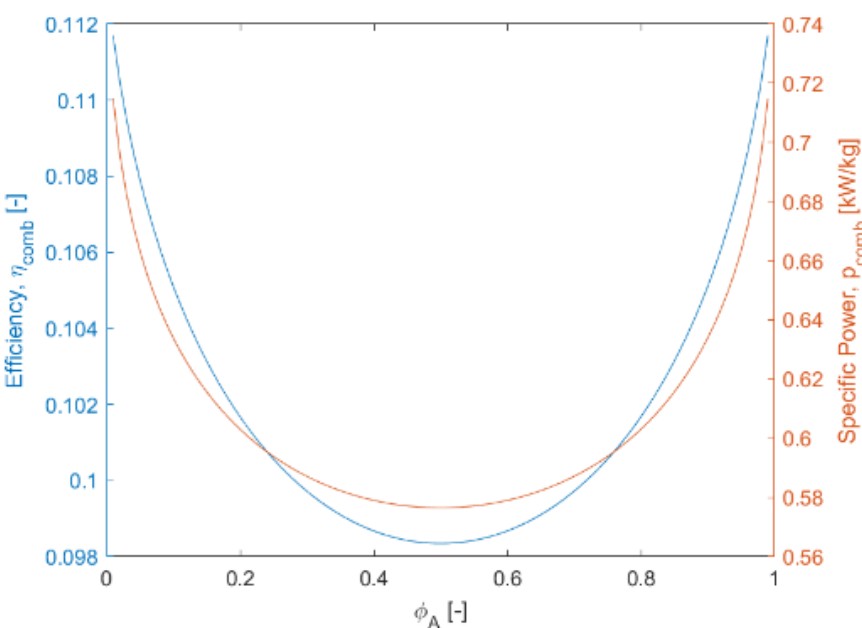

**Figure 10.** Combined mass and efficiency of two-branch system with non-constant component parameters.

Assuming that, rather than two branches, N branches are present, and that power is split evenly between these N branches, then it may be shown that the efficiency and specific power are given by:

$$\eta_{comb} = \alpha \left( \frac{P_o}{\eta_1} \right)^\beta \left( \frac{\eta_1}{N^\beta} \right) \tag{16}$$

$$m_{comb} = \frac{P_o}{\eta_1} \left[ \frac{1}{p_1} + \frac{N^\delta}{\gamma \left( \frac{P_o}{\eta_1} \right)^\delta} \right] \tag{17}$$

Figure 11 shows how the combined system mass, equivalent system specific power and system efficiency varies with the number of branches and the power requirement. It is interesting to note that efficiency decreases with more channels. This is in direct competition with reliability requirements that prefer multiple channels to ensure redundancy [20]. Nevertheless, it is also interesting to note that systems with more total power are more efficient and have a higher specific power.

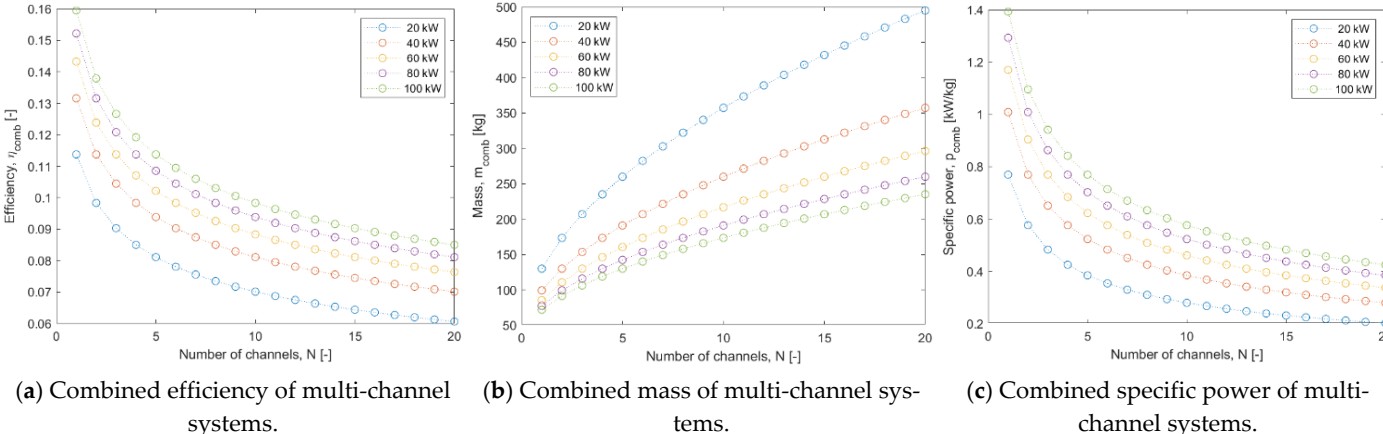

(**a**) Combined efficiency of multi-channel systems.

(**b**) Combined mass of multi-channel systems.

(**c**) Combined specific power of multi-channel systems.

**Figure 11.** System parameters of N-branch system with non-constant component parameters.

## 3. Sizing Procedure

The sizing procedure aims to determine the total, empty, fuel, and battery mass from mission requirements. Similar to the methodologies by Roskam [12], Raymer [11], and Riboldi and Gualdoni [14,15], the total mass of the aircraft $W_{tot}$ may be defined to be composed of:

$$W_{tot} = W_e + W_{pl} + W_{bat} + W_{fuel} \tag{18}$$

where $W_e$ is the empty mass, $W_{pl}$ is the mass of the payload including passengers and crew, $W_{bat}$ is the mass of the battery and $W_{fuel}$ is the mass of fuel. Payload is specified by the mission, the battery and fuel masses are determined from mission analysis, and finally, a statistical relationship may be used to find empty and total mass. These masses may be further specified if desired. For example, motor mass $W_m$ or powertrain mass $W_{active}$ may be treated separately from empty mass even though they have been included for current purposes.

### 3.1. Energy Modelling by Mission Analysis

Determining the battery and fuel mass is achieved by considering a known relationship between battery or fuel mass and energy, specific energy for batteries and specific fuel consumption (SFC) for combustion engines. Table 4 shows these values for batteries, diesel engines, and turboshaft engines found in the literature. It is expected that radically different but still unproven battery concepts such as Li-Air may reach specific energy values as high as 750 Wh/kg to 2000 Wh/kg [22]. Table 5 summarises the values found and has calculated the sample mean, median, and variance.

**Table 4.** Specific energy and specific fuel consumption found from literature.

| Component | Values | | | | | | | |
|---|---|---|---|---|---|---|---|---|
| Turboshaft SFC | 0.42 [30] | 0.31 [10] | | | | | | |
| Diesel SFC | 0.21 [30] | | | | | | | |
| Battery specific energy | 0.10 [31] | 0.15 [31] | 0.10 [6] | 0.25 [6] | 0.20 [13] | 0.14 [33] | 0.10 [22] | 0.20 [22] |
| | 0.14 [1] | 0.25 [10] | 0.03 [32] | 0.30 [32] | | | | |
| Near-term | 0.40 [31] | 0.40 [13] | 0.50 [25] | 0.20 [1] | 0.30 [1] | 0.50 [22] | | |
| Long-term | 0.90 [36] | 1.30 [36] | 0.75 [13] | 1.50 [13] | 0.75 [35] | 0.75 [22] | 2.00 [22] | 1.87 [26] |
| | 1.00 [25] | 0.30 [1] | 0.50 [1] | 0.70 [27] | | | | |

**Table 5.** Compiled values of state-of-the-art specific energy, e [kWh/kg] and Specific Fuel Consumption, SFC [kg/kWh], of energy sources.

| Component | Minimum | Maximum | Sample Mean | Sample Median | Sample Variance ($\times 10^{-3}$) |
|---|---|---|---|---|---|
| Turboshaft SFC (200 kW) | 0.31 | 0.42 | 0.37 | 0.37 | 5.62 |
| Diesel SFC | 0.21 | 0.21 | 0.21 | 0.21 | - |
| Battery specific energy, $e_{bat}$ | 0.03 | 0.30 | 0.16 | 0.15 | 6.16 |
| Near-term | 0.20 | 0.50 | 0.38 | 0.40 | 13.67 |
| Long-term | 0.30 | 2.00 | 1.03 | 0.83 | 283.3 |

Battery mass and fuel mass may be calculated using:

$$W_{bat} = E_{bat} \cdot e_{bat} \tag{19}$$

$$W_{fuel} = \frac{E_{fuel}}{e_{fuel}} = \eta_{ICE} \cdot E_{fuel} \cdot SFC \tag{20}$$

where $E_{bat}$ is the total energy the battery supplied during the entire mission, $e_{bat}$ is the specific energy of the battery, $E_{fuel}$ is the energy supplied from fuel during the entire

mission, $e_{fuel}$ is the specific energy of the fuel used, $SFC$ is the specific fuel consumption of the engine used, and $\eta_{ICE}$ is the efficiency of the engine used. Note that either $SFC$ of the engine or $e_{fuel}$ may be used where available, where $SFC = \frac{1}{\eta_{ICE} \cdot e_{fuel}}$.

Values for $E_{bat}$ and $E_{fuel}$ are derived from the aircraft mission. Conventionally, the energy requirements of a mission are determined for each phase of flight and expressed as a fuel fraction [11,12]. Unlike conventional fuels, batteries do not change mass during flight [14], and energy takes on multiple forms for hybrid-electric platforms. Therefore, rather than fuel fractions, the energy supplied by the battery and from fuel is expressed directly as $E_{bat}^{phase}$ and $E_{fuel}^{phase}$ for each phase of flight. Note that Riboldi and Gualdoni [14] have used battery specific power, $p_{bat}$, as well as battery specific energy, $e_{bat}$, for sizing the battery pack. For current analysis, it was assumed that $e_{bat}$ would be sufficient since $e_{bat}$ is far lower than $p_{bat}$, hence a more restricting constraint.

The parameter $H_e^{phase}$ specifies the hybridisation of energy and determines how much of the energy used during each phase must be supplied by the battery:

$$E_{bat}^{phase} = H_e^{phase} \frac{E_{phase}}{\eta_{sys}} \tag{21}$$

$$E_{fuel}^{phase} = \left(1 - H_e^{phase}\right) \frac{E^{phase}}{\eta_{sys}} \tag{22}$$

where $E^{phase}$ is the amount of energy required for that phase of flight, and $\eta_{sys}$ is the powertrain system efficiency—which takes into account energy losses over the powertrain and was determined previously through powertrain modelling. $H_e = 1$ for an all-electric platform. $H_e^{phase}$ is assumed to be constant for the entire phase.

Lastly, the energy required per flight phase may be determined by considering the power requirements per phase, $P^{phase}$, which is the amount of power required to maintain the flight condition and is found through aerodynamic analysis. This may be integrated over time to determine the total energy required per phase; however, in the current analysis, $P^{phase}$ was assumed constant:

$$E^{phase} = \int_{t_0}^{t_{end}} P^{phase}(t)dt = P^{phase} \cdot T^{phase} \tag{23}$$

Figure 12 shows two generic mission profiles for fixed wing as well as VTOL configurations. For fixed-wing case studies, just three mission phases: climb, cruise, and loiter were considered. Take-off and landing are assumed to require a negligible amount of energy compared to the other three phases. Furthermore, since the energy required for take-off and landing cannot be determined through a static flight equilibrium, but is often based on empirical data, these two phases have been disregarded from the analysis.

For VTOL configurations, four phases are considered: hover (take-off), climb, cruise, and hover (landing), where hover (take-off) and hover (landing) are considered aerodynamically the same. The point of transition, where the aircraft changes from hover into forward flight, and from forward flight into hover [37], was assumed to be negligible. Hover, climb, and cruise were similarly assumed to be the main contributors to the energy required for flight and benefit from straightforward aerodynamic analysis.

Since climb time, $T^{climb}$, cruise time, $T^{cruise}$, and hover time, $T^{hover}$ are not stated explicitly in the mission requirements, they have been defined as follows:

$$T^{climb} = \frac{h_{cruise}}{RC} \tag{24}$$

$$T^{cruise} = \frac{R}{V^{cruise}} \tag{25}$$

$$T^{hover} = \frac{h_{hover}}{V^{hover}} \tag{26}$$

Note, VTOL concepts may be realised through different configurations including Vectored Thrust, Lift + Cruise, and wingless configurations [6]. The case studies discussed during the research concern Vectored thrust and Lift + Cruise configurations and are assumed to have the same mission profile.

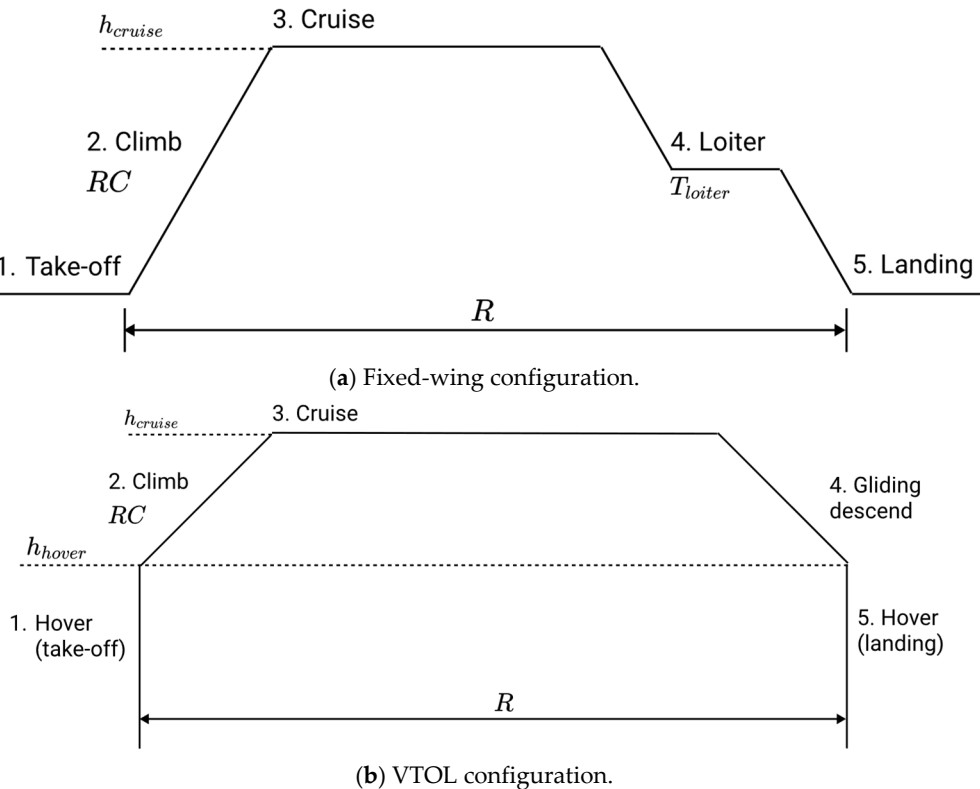

**Figure 12.** Generic mission profiles.

### 3.2. Aerodynamic Modelling

Generally speaking, aerodynamic modelling requires knowledge of the geometric properties of the aircraft as well as the flight conditions of each phase of flight as specified in the mission. Aerodynamic modelling determines the power required for each phase of flight and hence is vital. Furthermore, accurate aerodynamic modelling is also crucial to the point performance analysis as it describes the flight conditions of the performance requirements, making it possible to constrain the viable design space [12].

Riboldi and Gualdoni [14] have suggested that calculating the power loading required for each phase of flight, $P_{req}/W_{tot}$, can be achieved by analysing the forces on the aircraft in equilibrium flight for each phase [14]. Therein, they have used the procedure by Roskam [12] for finding the lift coefficient, $C_L$, and drag coefficient, $C_d$:

$$C_L = \frac{2}{\rho V^2}\left(\frac{W}{S}\right) \tag{27}$$

$$C_d = C_{d,0} + kC_L^2 \tag{28}$$

Thus, the power loading of the climb, cruise, and loiter phases may be derived to be [14]:

$$\frac{P_{req}^{climb}}{W_{tot}} = gRC + \frac{1}{2}C_d^{climb}\rho^{climb}V^{climb3}\left(\frac{W_{tot}}{S}\right)^{-1} \tag{29}$$

$$\frac{P_{req}^{cruise}}{W_{tot}} = \frac{1}{2} C_d^{cruise} \rho^{cruise} V^{cruise3} \left( \frac{W_{tot}}{S} \right)^{-1} \tag{30}$$

$$\frac{P_{req}^{loiter}}{W_{tot}} = \frac{1}{2} C_d^{loiter} \rho^{loiter} V^{loiter3} \left( \frac{W_{tot}}{S} \right)^{-1} \tag{31}$$

where $\rho$ is the air density during each phase, $C_d$ is the drag coefficient during each phase, $V$ is the airspeed during each phase, $g$ is the gravitational constant, $RC$ is the rate-of-climb, and $W_{tot}/S$ is the wing loading obtained from point-performance analysis.

As suggested by Bacchini and Cestino [6], disk-actuator theory may be applied to the hover phase to find the required power loading:

$$\frac{P_{req}^{hover}}{W_{tot}} = \frac{k_{int}}{2} \sqrt{\frac{g W_{tot}}{\rho_{hover} A_{disk}}} \tag{32}$$

Thus, these four equations are assumed to be a sufficient aerodynamic analysis for a conceptual stage in the design process, enabling the sizing process to be carried out.

The aerodynamic analysis used has been rudimentary since it is not the main focus of the research. For example, De Vries et al. [38] proposed a hybrid electric aircraft with distributed propulsion and duly noted that the selection of the powertrain will impact the aerodynamic behaviour of the system. Similarly, when considering configurations with a significant impact on aerodynamic behaviour, a surrogate model describing the relationship between the powertrain and the aerodynamic forces would be required for an improved overall aerodynamic model.

Further improvements to the aerodynamic model may concern how power loading changes as the weight of the aircraft decreases due to fuel burn. In the conventional design process, this was accounted for through the use of fuel fractions [12]. For the current aerodynamic model, power loading was assumed to be constant during the entire mission, contrary to it being likely to decrease over time as weight decreases. Hence, an improved aerodynamic model may include a way to model weight decrease over time. This would also mean that the effects of hybridisation become more pronounced since the battery does not lose weight as conventional powertrains do due to fuel burn, meaning that the relative size of the two types of powertrains becomes more important for power loading, aerodynamics and finally sizing. Other approaches to aerodynamic modelling exist as well, such as the use of lift-to-drag ratios rather than drag polars as performed by Melo [39] for the lifecycle analysis of eVTOL vehicles.

### 3.3. Mass Regression Relationship

Finally, Roskam [12] has proposed a statistical relationship between $W_{tot}$ and $W_e$ in the case of conventional aircraft:

$$\log(W_{tot}) = A \log(W_e) + B \tag{33}$$

Riboldi and [14] have shown that the relationship holds when applied to the case of electric aircraft. From a literature review, 25 aircraft, including hybrid and all-electric as well as fixed-wing and VTOL configurations, have been plotted in and fitted to the aforementioned statistical relationship (Figure 13). It shows that the relationship not only holds for electrical, VTOL, fixed wing, and hybrid configurations, but also that all configurations follow the same relationship with only slight variation in the case of VTOL. Although the regression is large which makes high accuracy in mass estimation unlikely, the identified relationship is adequate for conceptual design purposes.

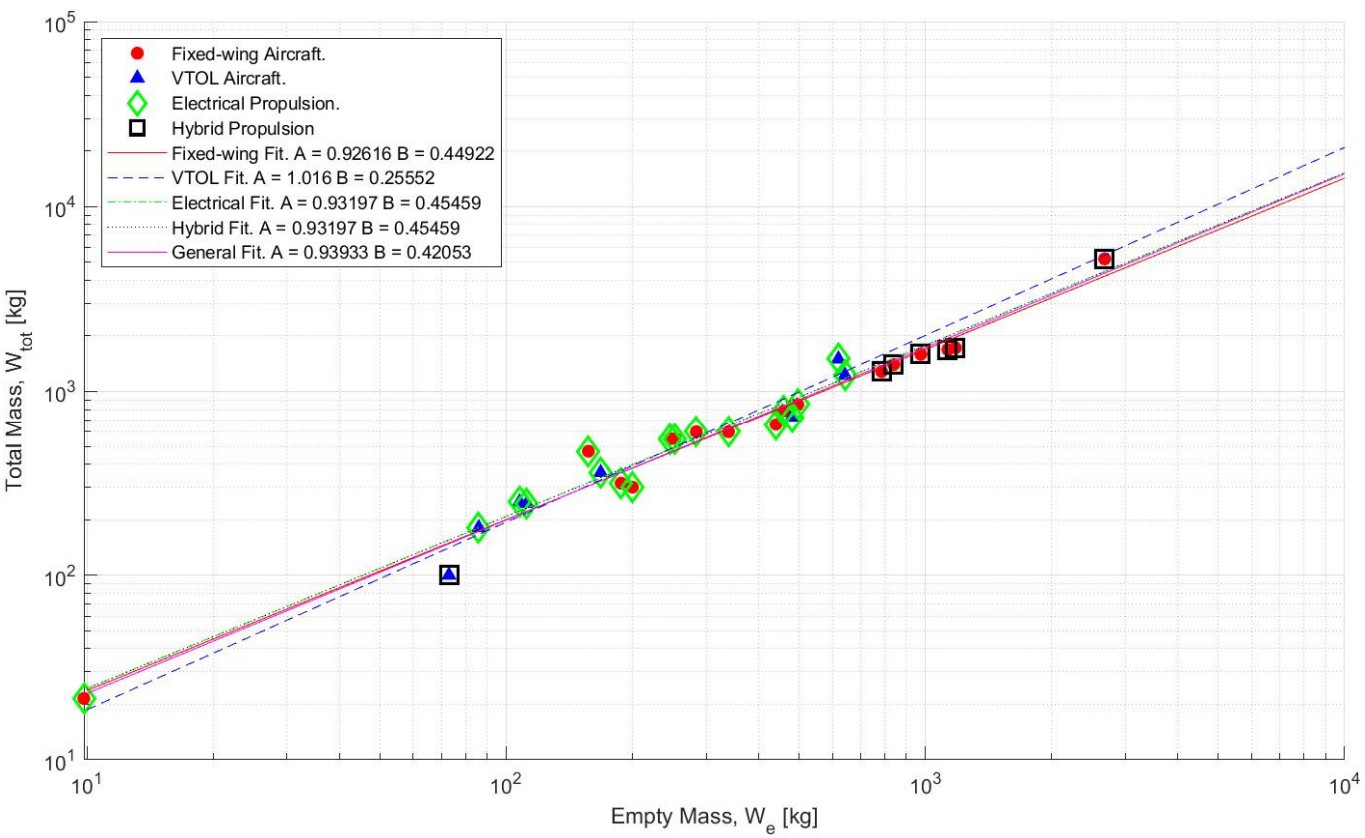

**Figure 13.** Mass relationship from literature [6–8,10,14,40–48].

It also points to an interesting idea about both mission and type of aircraft. More demanding missions—corresponding to more payload and/or greater ranges—are more likely to be hybrid-powered rather than all-electric. This is in line with analysis of the present industry, where all-electric configurations are only viable for smaller missions [1,4].

Since $W_{bat}$ and $W_{fuel}$ have been determined from the mission, and $W_{pl}$ is given, it is possible to perform a regression on the relationship in Figure 13. By basing an initial guess of total mass $W_{tot,guess}$ on previous aircraft with similar missions, it will be possible to use the relationship and converge on suitable values for $W_{tot}$, $W_e$, $W_{bat}$, and $W_{fuel}$. Not all aircraft will converge however, and this indicates that the mission specified cannot be fulfilled with current levels of technology. For example, a mission may specify an amount of energy that requires batteries bigger than any aircraft concept could carry, indicating that either the mission must be reduced, or the hybridisation must be decreased to allow more energy to come from conventional fuels.

## 4. Model Validation

The methodology described in the previous two sections was implemented in MATLAB and compared against data available from the literature. The implementation of the powertrain model was verified by testing it against trivial examples, for example by constructing a series or parallel connection consisting of just a single element. Another method to verify the distribution of mass and power is consistency, as the total mass must be the sum of the mass of each element. Lastly, the implementation may be manually verified using the equations themselves.

A total of five case studies have been found in the literature which are representative of a wide range of missions and aircraft designs: General Aviation [10], Glider [14], Unmanned Logistics [41], Urban Mobility (5 passengers) [6], and Urban Mobility (10 passengers) [49], which are based on the profiles given by Figure 12a,b; it includes fixed wing as well as VTOL configurations. The power loading has already been given for these case studies

and has been applied for powertrain modelling to determine the active mass, efficiency, and distribution.

A sensitivity analysis was carried out for the powertrain modelling and sizing procedures of two case studies. An analytical approach was taken for powertrain modelling whereas an experimental approach was taken for the sizing procedure. These analyses offer possible explanations for the discrepancy between model and case studies as they identify the contribution of variables like specific energy or mission length to the end results. This points towards avenues of improvement as the analyses have identified which variables have the most impact on the results and are most interesting for further work.

### 4.1. Case Studies

Table 6 specifies the missions of each of the fixed-wing case studies and VTOL case studies; where unavailable from literature, mission parameters have been estimated. Hybrid and all-electric powertrains are discussed.

**Table 6.** Mission specifications.

| | General Aviation | Glider | Logistics | Urban (5 PAX) | Urban (10 PAX) |
|---|---|---|---|---|---|
| Zero- lift drag coefficient $C_{d,0}$ | 0.0254 [10] | 0.011 [14] | 0.015 | 0.0163 [6] | 0.015 |
| Induced drag factor, $k$ | 0.0402 [10] | 0.0128 [14] | 0.0265 | 0.0580 [6] | 0.029 |
| Air density, $\rho$ . [kg/m$^2$] | | 0.909 | 1.169 | 1.168 | 1.168 |
| Climb | 1.167 | | | | |
| Cruise | 1.112 | | | | |
| Loiter | 1.167 | | | | |
| Airspeed, $V$ [m/s] | | | | | |
| Climb | 40 | 24.7 [14] | 40 | 50 | 50 [49] |
| Cruise | 90 [10] | 46.3 [14] | 45.8 [41] | 80.6 | 67 [49] |
| Loiter | 45 | $0.9V_{cruise}$ | | | |
| Hover (take-off) | | | 3 | 5 | 5 [49] |
| Hover (landing) | | | 3 | 5 | 1.5 [49] |
| Hybridisation, $H_e$, $H_p$ | | | | | |
| Climb | 0% | 100% | 0% | 100% | 100% |
| Cruise | 0.25% | 100% | 0% | 100% | 100% |
| Loiter | 0% | 100% | | | |
| Hover (take-off) | | | 100% | 100% | 100% |
| Hover (landing) | | | 100% | 100% | 100% |
| Battery specific energy, $e_{bat}$ [kWh/kg] | 0.25 [10] | 0.15 | 0.15 | 0.15 | 0.15 |
| Interference factor, $k_{int}$ | | | $\sqrt{2}$ | $\sqrt{2}$ | $\sqrt{2}$ |
| Hover altitude, $h_{hover}$ [m] | | | 2000 | 75 | 150 [49] |
| Disk area, $A_{disk}$ [m$^2$] | | | $8 \cdot \pi(1.12)^2$ [12] | $36 \cdot \pi(0.28)^2$ [6] | $12 \cdot \pi(0.75)^2$ [49] |
| Cruise altitude, $h_{cruise}$ [km] | 1.0 | 3.0 [14] | 5.0 | 0.5 | 0.5 [49] |
| Rate of climb, RC [m/s] | 5 [10] | 2.02 [14] | 3 | 8 | 8 [49] |
| Range, R [km] | 1150 [10] | 300 [14] | 300 | 245 | 200 [49] |
| Loiter time, $T_{loiter}$ [min] | 45 [10] | 15 [14] | | | |

For the hybrid-electric case studies, an input parameter, $H_p$, has been defined which specifies the hybridisation of power and determines how much power comes from the battery and how much comes from fuel:

$$P_{bat} = H_p P_{in} \tag{34}$$

$$P_{fuel} = (1 - H_p)P_{in} \tag{35}$$

This parameter determines the distribution of power within the powertrains and thus is used for the sizing of powertrain components. It corresponds to the output parameter, $\phi$, which is the ratio of the output power from one component (in this case from the battery, $P_{bat}$) to the total output power (in this case, the total power provided by battery and fuel,

$P_{in}$). This parameter is not the same as the energy hybridisation parameter, $H_e$, specified previously, which is associated with energy modelling, not powertrain modelling. $H_p$ is also assumed to be constant for the entire mission and to be decoupled from $H_e$. In actuality, $H_p$ may be calculated based on $H_e$ and vice versa. Nevertheless, this is outside the scope of this paper, and for simplicity, they have been assumed to be independent and a reasonable value of $H_p$ was guessed based on $H_e$.

Table 7 shows the results of the sizing procedure applied to the five aircraft. Some aircraft show good agreement with the actual design whereas some other case studies had fuel and battery masses that were lower than expected such as the General Aviation and Logistics missions. Interestingly, the Urban (5 PAX) case study was predicted a significantly higher empty mass ($W_e^{model} = 931$ kg; $W_e^{actual} = 236$ kg) and significantly smaller battery ($W_{bat}^{model} = 198$ kg; $W_{bat}^{actual} = 900$ kg) even if the total mass remained fairly accurate ($W_{tot}^{model} = 1692$ kg; $W_{tot}^{actual} = 1700$ kg). It is interesting to note that the case study does not seem viable to begin with and deviates significantly from the established relationship in Figure 13.

**Table 7.** Sizing and powertrain analysis results.

| Case Study | General Aviation [10] | | Glider [14] | | Logistics [41] | | Urban (5 PAX) [6] | | Urban (10 PAX [40]) | |
|---|---|---|---|---|---|---|---|---|---|---|
| **Specified parameters** | | | | | | | | | | |
| Given power loading | 6.1 kg/kW | | 20.5 kg/kW | | 4.32 kg/kW | | 9.1 kg/kW | | 3.8 kg/kW | |
| Given wing loading | 135 kg/m$^2$ | | 61 kg/m$^2$ | | 73.9 kg/m$^2$ | | 472 kg/m$^2$ | | 137 kg/m$^2$ | |
| Payload mass | 380 kg | | 150 kg | | 300 kg | | 564 kg | | 1000 kg | |
| **Sizing procedure results** | | | | | | | | | | |
| | Model | Actual | Model | Actual | Model | Actual | Model | Actual | Model | Actual |
| Battery mass [kg] | 32.5 | 33.3 | 171 | 241 | 136 | 576 | 198 | 900 | 661 | 1218 |
| Fuel mass [kg] | 198.3 | 385.9 | | | 9.16 | 65 | | | | |
| Empty mass [kg] | 722 | 1556 | 352 | 402 | 503 | 759 | 931 | 236 | 1935 | 1357 |
| Total mass [kg] | 1333 | 2355 | 672 | 793 | 948 | 1700 | 1682 | 1700 | 3595 | 3575 |
| Accuracy of prediction. | 56.6% | | 84.7% | | 55.8% | | 99.5% | | 101% | |
| **Powertrain modelling results** | | | | | | | | | | |
| Powertrain | Series-hybrid | | All-electric | | Parallel hybrid | | All-electric | | All-electric | |
| Deliverable power | 218.5 kW | | 32.8 kW | | 219.5 kW | | 86 kW | | 946.2 kW | |
| Modelled active mass | 302.3 kg | | 14.1 kg | | 109 kg | | 80.1 kg | | 408 kg | |
| Modelled system efficiency | 19.3% | | 68.6% | | 33.2% | | 68.6% | | 68.6% | |
| | Model | Actual | Model | Actual | Model | Actual | Model | Actual | Model | Actual |
| Modelled active mass to total mass | 22.7% | 12.8% | 2.1% | 1.8% | 11.5% | 6.4% | 4.7% | 4.7% | 11.3% | 11.4% |
| Modelled active mass to empty mass | 41.9% | 19.4% | 4.0% | 3.5% | 21.7% | 14.4% | 8.6% | 33.9% | 21.1% | 30.0% |

The method seems to be more accurate for all-electric cases compared to hybrid aircraft. This may correlate with limitations of the aerodynamic model, which assumes that aircraft mass remains the same for all phases. In reality, aircraft mass decreases due to burning fuel. This complicates the aerodynamic model and leads to changes in the power requirements for each phase, this changes the requires battery size and fuel capacity and may influence the final results. From mission analysis, it is clear that the powertrain is capable of generating more power than required by the missions. This suggests that the required power loading as determined by the mission analysis is just a lower bound. It does not take into account further requirements coming up during performance analysis or otherwise. Furthermore, no case study powertrain is undersized, i.e., Providing less power than required by the missions they were assigned.

Table 7 shows the general results of powertrain modelling calculated using the specified power loading—rather than the power loading derived from mission analysis. Figure 14 shows the mass and power distribution of the General Aviation case study. Note that system efficiency includes battery and combustion engine efficiencies. The pre-

dicted active masses ranges from 3.5% to 33.9% of the actual empty mass—reflecting the differences in performance requirements. As such, the motor-glider case study has the smallest powertrain.

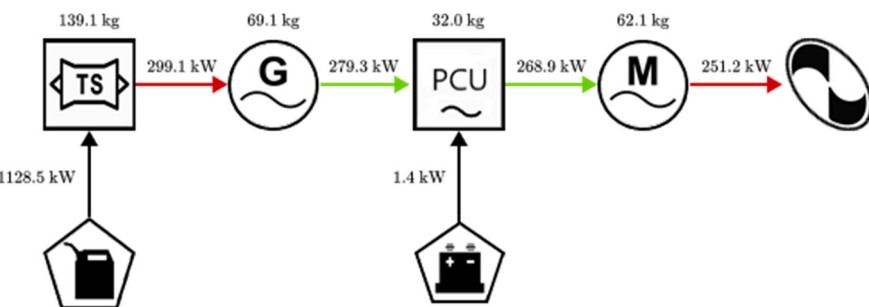

**Figure 14.** Power and mass distribution of General Aviation case study.

However, it may be noted that the motor for the motor-glider mission is smaller than predicted by Riboldi and Gualdoni, $W_m = 23$ kg [14], which is likely caused by the different non-linear model Riboldi and Gualdoni used.

### 4.2. Sensitivity Analysis

The quality of these design solutions may be assessed through a sensitivity analysis of the powertrain model and sizing procedure. Starting with the powertrain, an analytical method for determining the sensitivity of these variables is preferred because analytical forms for system efficiency and system specific power are known, and the system is influenced by a great number of parameters. Thus, a concise method for determining the sensitivity of the system to each parameter would be using the variance equation [50],

$$\sigma_f^2 = \sum_{i=1}^{N} \left( \frac{\partial f}{\partial x_i} \right)^2 \sigma_{x_i}^2 \tag{36}$$

to calculate the contribution of a parameter to the total variance of the system. It assumes that the parameters are Normally distributed and independent.

For simplicity, only the system efficiency and system specific power have been considered for this project, which assumes that there is a high degree of certainty in the amount of power required from the system. However, if desired a similar procedure may be applied to total powertrain mass, which is capable of directly taking into account uncertainty of the power draw of the powertrain and assessing its importance.

Since powertrains are represented using parallel and series combinatory rules, different expressions must be derived describing each connection. These may be combined to represent any powertrain and calculate its total variance.

Figure 15 shows the contribution of each component of an all-electric case study (Figure 3d). The variance used for these calculations were taken to be the sample variances of Tables 2, 3 and 5. The variance of system efficiency is $\sigma_{\eta_{sys}} = 1.63 \cdot 10^{-4}$, and the variance of system specific power is $\sigma_{p_{sys}} = 7.33 \cdot 10^2$.

Assessing the sensitivity of the sizing procedure requires a different methodology since the sizing procedure is an iterative process and cannot be expressed analytically; the variance equation cannot be applied. Furthermore, it must be noted that there are significant differences between each case study since sizing is strongly influenced by the mission. Taking advantage of the low computational cost of the methodology, it is possible to assess sensitivity by perturbing each parameter to the sizing procedure and assessing its impact on the results.

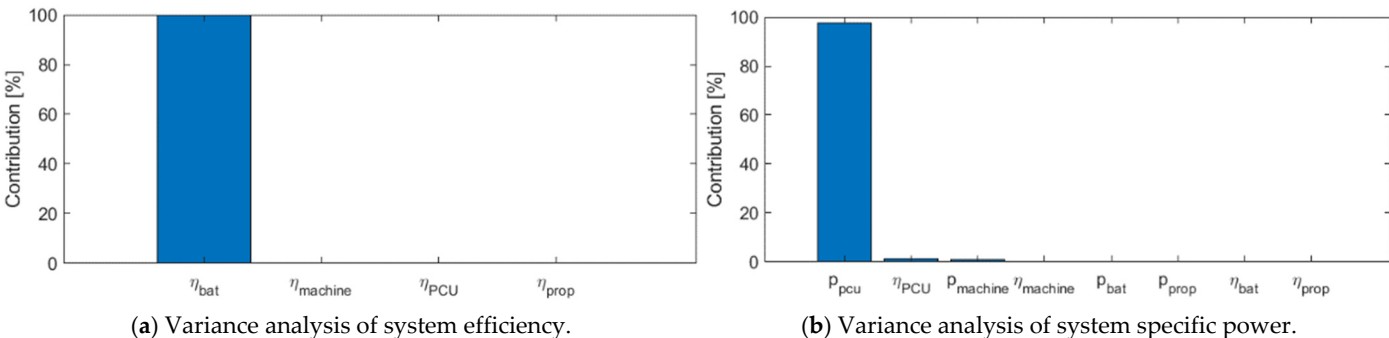

(**a**) Variance analysis of system efficiency.      (**b**) Variance analysis of system specific power.

**Figure 15.** Variance analysis of all-electric powertrain.

Figure 16 shows the full sensitivity analyses carried out for the sizing of two case studies: Hybrid General Aviation and All-electric Urban Mobility (5 PAX). Considering the results, it is interesting to note that variation in power (Figure 16a,h) and flight time (Figure 16b,i) will yield the same results since both linearly influence the energy demand by the relationship: $E = PT$. Because energy was modelled in this particular manner, a variation in flight time is the same as a variation in power since both control energy demand.

It is also important to note that for all parameters of these two case studies, total and empty mass are more robust to parameter variation compared to fuel and battery mass. The high robustness to changes in specific energy is especially noteworthy and means that the current design is protected against future changes or current uncertainty surrounding state-of-the-art battery technology.

Looking at the hybrid configuration in particular, it is also evident that fuel mass is more sensitive than battery mass, which may possibly be caused by the low hybridisation of the case study considered. This sensitivity analysis also explains why modelled battery and fuel masses tend to deviate more from their actual values compared to empty or total masses since these values are more sensitive to variations in specific fuel consumption or battery specific energy.

Returning to the discrepancy between predicted and actual mass for the two case studies, it is now clear that there may be multiple confounding variables either originating from powertrain analysis—as indicated by the high sensitivity to variation in efficiency (Figure 16f,k)—or aerodynamic analysis—as indicated by power (Figure 16a,h), time (Figure 16b,i), and wing-loading (Figure 16g,l). This means that both powertrain modelling and an improved aerodynamic analysis are vital for the sizing process to be accurate. As an example, for the General Aviation case study, an increase of 20% in the power required (Figure 16a) and 20% increase in SFC (Figure 16e) would already lead to a more accurate prediction of total aircraft mass of over 80% compared to the actual total mass. Attention may be paid to the comparative robustness to variation of the Urban (5 PAX) case study compared to the General Aviation case study, which may explain the higher discrepancy in sizing of the latter.

Comparing the two methods used for sensitivity analysis, the analytical method utilising the variance equation excels in identifying the relative contribution of each component to the total variance of the system. It allows one to identify the most important parameters and components of the powertrain. This enables further analysis since it narrows down the sensitivity to a select few parameters and allows for these to be considered more closely if required.

The sensitivity analysis of the sizing procedure on the other hand is non-linear and represents deviations from the chosen design point. Although capable of clearly describing how total mass changes as parameters change, it is also limited to the chosen design point. These results cannot be generalised to all aircraft and must be repeated for each configuration. If too many parameters are present, the analysis will also be very computationally expensive.

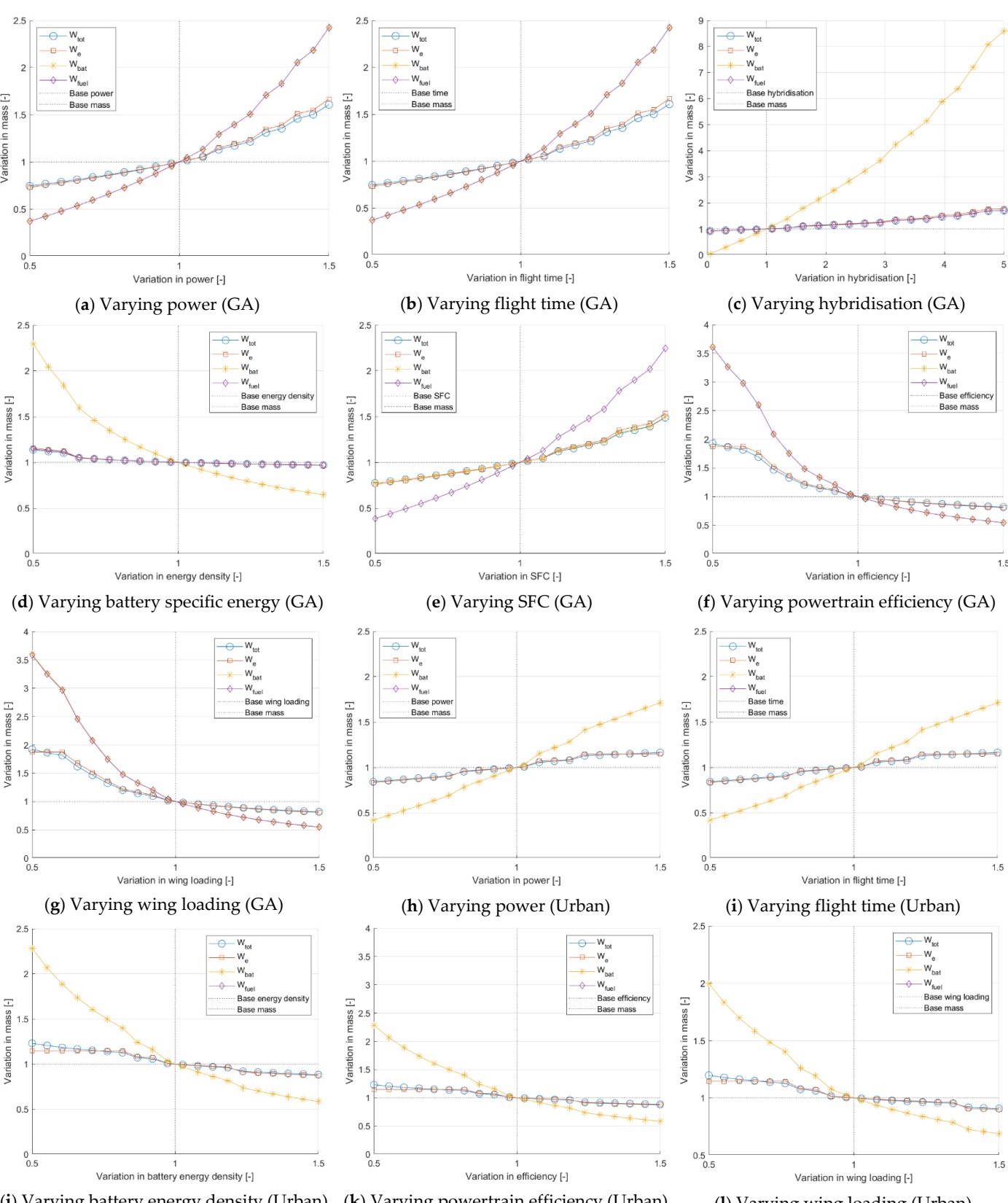

(**a**) Varying power (GA)

(**b**) Varying flight time (GA)

(**c**) Varying hybridisation (GA)

(**d**) Varying battery specific energy (GA)

(**e**) Varying SFC (GA)

(**f**) Varying powertrain efficiency (GA)

(**g**) Varying wing loading (GA)

(**h**) Varying power (Urban)

(**i**) Varying flight time (Urban)

(**j**) Varying battery energy density (Urban)

(**k**) Varying powertrain efficiency (Urban)

(**l**) Varying wing loading (Urban)

**Figure 16.** Sensitivity analysis of sizing procedure for General Aviation (GA) and Urban VTOL (5 PAX).

## 5. Conclusions and Further Work

This research has conducted a literature review of existing design methodologies and presented state-of-the-art values for powertrain components. A four-part methodology was constructed, and from literature, five case studies—representing a wide range of missions and configurations—were found on which the methodology was executed. Validation has demonstrated that the methodology can deliver accurate predictions of total and powertrain mass. Finally, a sensitivity analysis was performed, exposing the limitations of the sizing procedure and powertrain modelling.

Although the method is limited by some major assumptions, including the limited availability and, therefore, confidence in the data, it is versatile enough for future work to expand on and include emerging certification requirements on all-electric and hybrid electric as constraints on their conceptual design. The powertrain model may be expanded by the further integration of more complex surrogate models such as a model of battery power as a function of state-of-charge. Similarly, other components may be added such as inverters, converters, and rectifiers rather than a single PCU unit or two-way power paths could be included as De Vries et al. have done [38]. These improvements would allow for more detailed comparisons to be made, achieving higher fidelity modelling, and hence improving the accuracy of the methodology.

Further work may also be done in exploring the input and output parameters, $\phi$, ($\phi = H_p$, in the case studies discussed) and $\psi$ as well as hybridisation of energy, $H_e$. For example, the relationships which relate the hybridisation of the powertrain to the hybridisation of energy specified in the mission may be derived. It could also include work on standardising the format of how a hybrid-electric platform is specified in top-level requirements. Further analysis may be extended by an investigation of the ideal hybridisation ratio and power distribution and how those change with top level requirements. This research is part of a broader integrated design process required for future aircraft design, and more attention may be paid to an improved aerodynamic analysis accompanied by a more detailed mission description for more accurate results. For example, improved aerodynamic analysis could consider the decreasing mass due to fuel burn and how that impacts the power requirements over time, which were assumed to be constant in this research. More detailed mission descriptions on the other hand could also include loiter for the VTOL missions. The strength of this methodology though lies in being able to accurately compare a great number of different powertrains and missions at low computational cost, providing the designer with the ability to quickly determine the size and distribution of a wide range of aircraft concepts.

**Author Contributions:** J.H. and J.B. developed the methodology, J.H. collated data and case studies and both authors contributed to the writing of the paper. All authors have read and agreed to the published version of the manuscript.

**Funding:** This research received no external funding.

**Institutional Review Board Statement:** Not applicable.

**Informed Consent Statement:** Not applicable.

**Data Availability Statement:** Not applicable.

**Conflicts of Interest:** The authors declare no conflict of interest.

## Nomenclature

| | |
|---|---|
| $P_k$ | Component element input power, series system input power. |
| $P_o$ | Component element output power, system output power |
| $\eta_k$ | Component element efficiency |
| $p_k$ | Component element specific power |
| $m\_k$ | Component element mass |
| $\eta_{sys}$ | System efficiency |
| $p_{sys}$ | System specific power |
| $m_{sys}$ | System mass |
| $\phi_k$ | Input parameter, ratio of component input power to total system input power |
| $\psi_k$ | Output parameter, ratio of component output power to total system output power. |
| $P_N$ | Parallel system input power. |
| $W_{tot}$ | Aircraft total mass |
| $W_e$ | Aircraft empty mass |
| $W_{pl}$ | Aircraft payload mass |
| $W_{bat}$ | Aircraft battery mass |
| $W_{fuel}$ | Aircraft fuel mass |
| $W_m$ | Aircraft motor mass |
| $W_{active}$ | Aircraft powertrain mass |
| $SFC$ | Specific fuel consumption |
| $e_{bat}$ | Battery specific energy |
| $E_{fuel}$ | Energy supplied from fuel during the entire mission. |
| $E_{bat}$ | Energy supplied by the battery during the entire mission |
| $\eta_{ICE}$ | Combustion engine efficiency |
| $E_{bat}^{phase}$ | Energy supplied by the battery during a given phase |
| $E_{fuel}^{phase}$ | Energy supplied from fuel during a given phase |
| $H_e^{phase}$ | Hybridisation of energy, ratio of energy supplied by the battery to the total energy required during a given phase |
| $P^{phase}$ | Power requirement for a given phase |
| $T^{phase}$ | Duration of a given phase |
| $h_{cruise}$ | Cruising altitude |
| $V_{cruise}$ | Cruising speed |
| $h_{hover}$ | Hover altitude |
| $V_{hover}$ | Hover speed |
| $R$ | Range of aircraft |
| $RC$ | Rate of climb |
| $C_L$ | Coefficient of lift |
| $\rho$ | Air density |
| $\frac{W}{S}$ | Wing loading |
| $C_d$ | Drag coefficient |
| $C_{d,0}$ | Zero-lift drag coefficient |
| $\frac{P}{W}$ | Power loading |
| $g$ | Gravitational constant |
| $A_{disk}$ | Disk area |
| $H_p$ | Hybridisation of power, ratio of power supplied by battery to total power supplied. |
| $P_{in}$ | Gross power required by system |
| $k$ | Induced drag factor |
| $k_{int}$ | Interference factor |
| $\sigma_f^2$ | System variance |
| $\sigma_{x_i}^2$ | Component variance |

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
