# Peer review of "Preliminary Sizing of Electric-Propulsion Powertrains for Concept Aircraft Designs"

_designs, 2022_

Round 1

Reviewer 1 Report

The paper provides a good overview of a mass estimation method for electric powertrains for conceptual aircraft design. The reference data for efficiency, specific power values is very well organized. The result section gives an insight on how well the mass estimations agree with the reference aircraft concepts, depending on the degree of hybridization and the configuration. The limitations of the current method as well as the development potentials are well described. Furthermore, I would like to address several points as written below.

Orthographic/layout comments:
- Figure 1: Larger font size please.
- Figure with "Definition of single component..." should be "Figure 6".
- Table 5: Please provide second figure behind the decimal (e.g. 0.31 instead of 0.3)
- Figure 12a: Please draw loiter as a horizontal flight phase which interrupts the descent.
- Lines 519 ff.: Maybe add the numbers (236 kg empty mass and 1700 kg total mass) so that the reader could grasp the context more quickly?
- Table 6:  "Induced drag factor" (instead of "induced drug factor")
            For hover (take-off) and hover (landing), please include the parameter and the unit (at first, I thought of 3 minutes of hover instead of 3 m/s ascent/descent rate).
- Line 584: Isn't the all-electric case shown in Figure 3d?

Technical comments:
- Figure 13: The scatter in the regression seems to be large (with deviations up to 50%), therefore high accuracies in the mass estimation is unlikely to be reached, especially if the specific aircraft configuration (e.g. tandem wing, tiltable wing etc.) is not taken into account. However, the regression seems to be adequate for conceptual aircraft design.
- Lines 521 ff.: How are the certification requirements for the investigated aircraft that is equivalent to the one-engine-out condition in CS25 (when one engine/rotor fails and the aircraft still has to be able to take-off and climb)? Because that condition causes engines of CS25 aircraft to be much more powerful (and thus heavier) compared to the ideal case. Maybe mention briefly in the future work section?
- Lines 530 ff.: If compared to the empty masses, the estimated relative active masses do not seem to be accurate (e.g. 8.6% instead of 33.9% on the Urban 5 PAX configuration). If compared to the total masses however, the estimations seem to have a higher accuracy, especially for the full-electric aircraft concepts. Is the latter meant by the concerning sentence?
- Table 7 (for own understanding): The estimations of total mass seem to have good accuracy for fully battery-powered aircraft concepts, but the deviations are larger for aircraft with combustion engines. The deviation might correlate with the assumption that the aircraft mass does not change during the flight (which would cause errors on the fuel combusting aircraft concepts).
- Regarding mission profile: If the fixed-wing configurations have reserve for loiter, then the VTOL configuration should also allocate reserve energy for loiter or divert. This would increase the battery and/or fuel mass on the VTOL configurations. Is this feature planned in the future work?

Author Response

Dear Reviewer

We would like to thank you very much for your time reading the paper and providing positive comments but also areas for improvement of the paper. We have considered all points made carefully and presented our responses below in italic where required. I hope this provides sufficient response and we look forward to a positive outcome for the progress of the paper.

Your sincerely,

The Authors.

Review #1

The paper provides a good overview of a mass estimation method for electric powertrains for conceptual aircraft design. The reference data for efficiency, specific power values is very well organized. The result section gives an insight on how well the mass estimations agree with the reference aircraft concepts, depending on the degree of hybridization and the configuration. The limitations of the current method as well as the development potentials are well described. Furthermore, I would like to address several points as written below.

Orthographic/layout comments:

  • Figure 1: Larger font size please.
    • Figure has been made bigger.
  • Figure with "Definition of single component..." should be "Figure 6".
    • Changed to “Figure 6”.
  • Table 5: Please provide second figure behind the decimal (e.g. 0.31 instead of 0.3)
    • Second figure has been added.
  • Figure 12a: Please draw loiter as a horizontal flight phase which interrupts the descent.
    • Redrawn the loiter phase as a horizontal flight phase interrupting descent.
  • Lines 519 ff.: Maybe add the numbers (236 kg empty mass and 1700 kg total mass) so that the reader could grasp the context more quickly?
    • Added numbers to aid legibility and clarity.
  • Table 6:  "Induced drag factor" (instead of "induced drug factor")
    • Fixed typo.
  • Table 6: For hover (take-off) and hover (landing), please include the parameter and the unit (at first, I thought of 3 minutes of hover instead of 3 m/s ascent/descent rate).
    • Hover (take-off) and hover (landing) should have been indented. Under airspeed it indicates the airspeed of the hover phase in question. Under hybridisation it indicates the energy hybridisation of that phase. The indentation has been fixed.
  • Line 584: Isn't the all-electric case shown in Figure 3d?
    • Fixed typo, figure numbering was changed.

Technical comments:

  • Figure 13: The scatter in the regression seems to be large (with deviations up to 50%), therefore high accuracies in the mass estimation is unlikely to be reached, especially if the specific aircraft configuration (e.g. tandem wing, tiltable wing etc.) is not taken into account. However, the regression seems to be adequate for conceptual aircraft design
    • Comment has been included in Section 3.3.
  • Lines 521 ff.: How are the certification requirements for the investigated aircraft that is equivalent to the one-engine-out condition in CS25 (when one engine/rotor fails and the aircraft still has to be able to take-off and climb)? Because that condition causes engines of CS25 aircraft to be much more powerful (and thus heavier) compared to the ideal case. Maybe mention briefly in the future work section?
    • Agreed, this is now mentioned in the Conclusions section.
  • Lines 530 ff.: If compared to the empty masses, the estimated relative active masses do not seem to be accurate (e.g. 8.6% instead of 33.9% on the Urban 5 PAX configuration). If compared to the total masses however, the estimations seem to have a higher accuracy, especially for the full-electric aircraft concepts. Is the latter meant by the concerning sentence?
    • The “Model” and “Actual” columns in Table 7 do not reflect the modelled relative active to total mass predicted and the actual relative to active mass. The actual mass of the powertrains for the case studies is not available. Rather, Model indicates the ratio between modelled active mass to modelled total/empty mass and Actual indicates the ratio between modelled active mass to actual total/empty mass. The table formatting could have been misinterpreted.
    • It is only anecdotally determined to be accurate as no literature was consulted to make such a claim.
  • Table 7 (for own understanding): The estimations of total mass seem to have good accuracy for fully battery-powered aircraft concepts, but the deviations are larger for aircraft with combustion engines. The deviation might correlate with the assumption that the aircraft mass does not change during the flight (which would cause errors on the fuel combusting aircraft concepts).
    • Included in the paper (4. Model validation), further work to narrowing this discrepancy lies in creating a better aerodynamic model which takes into account mass lost due to fuel burn.
  • Regarding mission profile: If the fixed-wing configurations have reserve for loiter, then the VTOL configuration should also allocate reserve energy for loiter or divert. This would increase the battery and/or fuel mass on the VTOL configurations. Is this feature planned in the future work?
    • Included in Conclusions section as possible future work.

Reviewer 2 Report

This paper presents a simple and straightforward sizing methodology for the design of electric-propulsion aircraft. The method would be of interest for the community due to its fast implementation and applicability to a wide range of airframes.

Overall the paper is well written, and the capabilities and limitations of the methodology are acknowledged.

The only item of concern, that should be further addressed, is related to the discrepancies between the estimated and actual masses for some of the case studies. With the sensitivity analysis, the authors have provided a potential explanation for some of these discrepancies (e.g., the high sensitivity of fuel mass to SFC for the hybrid GA case, which will be used to further illustrate the comment). However, it seem interesting that the authors have not performed a further parametric study to assess the range of SFCs for which the predicted fuel mass is closer to the actual mass, and assess whether this range is realistic.

The reviewer believes it would be important, for any adoption of this model by the community, to verify that the discrepancies obtained for relatively classic geometries can be fully explained by lack of data and/or model sensitivity, and not by some fundamental issue in the methodology.

Author Response

Dear Reviewer,

We would like to thank you very much for your time reading the paper and providing positive comments but also areas for improvement of the paper. We have considered all points made carefully and presented our responses below in italic where required. I hope this provides sufficient response and we look forward to a positive outcome for the progress of the paper.

Your sincerely,

The Authors.

Review #2

This paper presents a simple and straightforward sizing methodology for the design of electric-propulsion aircraft. The method would be of interest for the community due to its fast implementation and applicability to a wide range of airframes. Overall the paper is well written, and the capabilities and limitations of the methodology are acknowledged. The only item of concern, that should be further addressed, is related to the discrepancies between the estimated and actual masses for some of the case studies. With the sensitivity analysis, the authors have provided a potential explanation for some of these discrepancies (e.g., the high sensitivity of fuel mass to SFC for the hybrid GA case, which will be used to further illustrate the comment).

  • However, it seem interesting that the authors have not performed a further parametric study to assess the range of SFCs for which the predicted fuel mass is closer to the actual mass,
    • A full sensitivity analysis has been included with appropriate references to the figures. These were kept in reserve due to page space initially, but actually provide further insight.
  • and assess whether this range is realistic.
    • Now stated that a 20% increase in Power required and 20% increase in SFC, leads to a more accurate total mass of 80%.

The reviewer believes it would be important, for any adoption of this model by the community, to verify that the discrepancies obtained for relatively classic geometries can be fully explained by lack of data and/or model sensitivity, and not by some fundamental issue in the methodology.

Reviewer 3 Report

The authors conducted a thorough literature review of existing design methodologies and presented state-of-the-art values for powertrain components. Five case studies are picked to represent a wide range of missions and configurations. The validation studies demonstrated that the methodology can deliver accurate predictions of total and powertrain mass. Also, a sensitivity analysis was performed. The authors also discussed the limitations of the sizing procedure and powertrain modeling and proposed future work.

The methodology is clear. The authors give a thorough description of all the components, which makes it easier to understand. Although there are limitations, the review thinks that this work still lays a solid foundation. The idea presented in the paper represents a sizeable effort to extend and develop the tool for quickly determining the size and distribution of a wide range of aircraft concepts.

A few suggestions.

In the Introduction section, the authors may want to specifically discuss the contribution and significance of this work.

There is no 1.2 in the Introduction section.

The formula and symbols are difficult to read in Figure 1, the authors should discuss the methods shown in Figure 1.

Author Response

Dear Reviewer,

We would like to thank you very much for your time reading the paper and providing positive comments but also areas for improvement of the paper. We have considered all points made carefully and presented our responses below in italic where required. I hope this provides sufficient response and we look forward to a positive outcome for the progress of the paper.

Your sincerely,

The Authors.

Review #3

The authors conducted a thorough literature review of existing design methodologies and presented state-of-the-art values for powertrain components. Five case studies are picked to represent a wide range of missions and configurations. The validation studies demonstrated that the methodology can deliver accurate predictions of total and powertrain mass. Also, a sensitivity analysis was performed. The authors also discussed the limitations of the sizing procedure and powertrain modeling and proposed future work.The methodology is clear. The authors give a thorough description of all the components, which makes it easier to understand. Although there are limitations, the review thinks that this work still lays a solid foundation. The idea presented in the paper represents a sizeable effort to extend and develop the tool for quickly determining the size and distribution of a wide range of aircraft concepts. A few suggestions.

  • In the Introduction section, the authors may want to specifically discuss the contribution and significance of this work.
    • Rather than in the introduction section, the contribution is confirmed more strongly now in the conclusions section.
  • There is no 1.2 in the Introduction section.
    • Typo fixed.
  • The formula and symbols are difficult to read in Figure 1
    • Figure has been enlarged.
  • The authors should discuss the methods shown in Figure 1.
    • A description of the four sub-procedures has been given in the introduction and relevant references provided for further insight to the different stages adopted.